# A generalizable and accessible approach to machine learning with global satellite imagery

Esther Rolf [1,2,9], Jonathan Proctor [3,9], Tamma Carleton [4,5,9], Ian Bolliger [2,6,9], Vaishaal Shankar[1,9], Miyabi Ishihara [2,7], Benjamin Recht[1] & Solomon Hsiang [2,5,8 ✉]

Combining satellite imagery with machine learning (SIML) has the potential to address global challenges by remotely estimating socioeconomic and environmental conditions in data-poor regions, yet the resource requirements of SIML limit its accessibility and use. We show that a single encoding of satellite imagery can generalize across diverse prediction tasks (e.g., forest cover, house price, road length). Our method achieves accuracy competitive with deep neural networks at orders of magnitude lower computational cost, scales globally, delivers label super-resolution predictions, and facilitates characterizations of uncertainty. Since image encodings are shared across tasks, they can be centrally computed and distributed to unlimited researchers, who need only fit a linear regression to their own ground truth data in order to achieve state-of-the-art SIML performance.

[1] Electrical Engineering & Computer Science Department, UC Berkeley, USA. [2] Global Policy Laboratory, Goldman School of Public Policy, UC Berkeley, USA. [3] Center for the Environment and Data Science Initiative, Harvard University, Cambridge, MA, USA. [4] Bren School of Environmental Science & Management, UC Santa Barbara, Santa Barbara, CA, USA. [5] National Bureau of Economic Research, Cambridge, MA, USA. [6] Rhodium Group, New York, USA. [7] Statistics Department, UC Berkeley, USA. [8] Centre for Economic Policy Research, London, UK. [9] These authors contributed equally: Esther Rolf, Jonathan Proctor, Tamma Carleton, Ian Bolliger, Vaishaal Shankar. ✉email: shsiang@berkeley.edu

Addressing complex global challenges—such as managing global climate changes, population movements, ecosystem transformations, or economic development—requires that many different researchers and decision-makers (hereafter, users) have access to reliable, large-scale observations of many variables simultaneously. Planet-scale ground-based monitoring systems are generally prohibitively costly for this purpose, but satellite imagery presents a viable alternative for gathering globally comprehensive data, with over 700 earth observation satellites currently in orbit[1]. Further, application of machine learning is proving to be an effective approach for transforming these vast quantities of unstructured imagery data into structured estimates of ground conditions. For example, combining satellite imagery and machine learning (SIML) has enabled better characterization of forest cover[2], land use[3], poverty rates[4] and population densities[5], thereby supporting research and decision-making. We refer to such prediction of an individual variable as a task. Demand for SIML-based estimates is growing, as indicated by the large number of private service-providers specializing in predicting one or a small number of these tasks.

The resource requirements for deploying SIML technologies, however, limit their accessibility and usage. Satellite-based measurements are particularly under-utilized in low-income contexts, where the technical capacity to implement SIML may be low, but where such measurements would likely convey the greatest benefit[6,7]. For example, government agencies in low-income settings might want to understand local waterway pollution, illegal land uses, or mass migrations. SIML, however, remains largely out of reach to these and other potential users because current approaches require a major resource-intensive enterprise, involving a combination of task-specific domain knowledge, remote sensing and engineering expertise, access to imagery, customization and tuning of sophisticated machine learning architectures, and large computational resources[8].

To remove these barriers, a new approach to SIML is needed that will enable non-experts to obtain state-of-the-art performance without using specialized computational resources or developing a complex prediction procedure. A one-time, task-agnostic encoding that transforms each satellite image into a vector of variables (hereafter, features) could enable such an approach by separating users from the costly manipulation of imagery. Such an unsupervised encoding might be particularly well suited for SIML problems, especially when contrasted with deep-learning approaches to SIML that use techniques originally developed for natural images (e.g., photos taken from handheld cameras). Inconsistency of many key factors in natural imagery, such as subject or camera perspective, require complex solutions that may be unnecessary for learning from satellite imagery. While prior work has sought an unsupervised encoding of satellite imagery[9–12], to date no single set of features has been shown to both achieve performance competitive with deep-learning methods across a variety of tasks and to scale globally.

Here we show that a single set of general purpose features can encode rich information in satellite images, performing well at predicting ground conditions across diverse tasks using only a linear regression implemented on a personal computer. We focus on the problem of predicting properties of small regions (e.g., average house price) at a single time period, using high-resolution daytime satellite imagery as the only input. We use this imagery to test whether a single embedding can generalize across tasks because it is globally available from the Google Static Maps API at fine resolution, is geo-rectified and pre-processed to remove cloud occlusions, and has been found to perform well in SIML applications (Supplementary Note 1.2)[4,13], though in principle other data sources could also be used[14]. We develop a simple yet high-performing system that is tailored to address the challenges and

opportunities specific to SIML applications, taking a fundamentally different approach from leading designs. We achieve large computational gains in model training and testing, relative to leading deep neural networks, through algorithmic simplifications that take advantage of the fact that satellite images are collected from a fixed distance and viewing angle and capture repeating patterns and objects. In addition, traditionally, hundreds or thousands of researchers use the same images to solve different and unrelated tasks (e.g., Fig. 1a). Our approach allows common sources of imagery to be converted into centralized sets of features that can be accessed by many researchers, each solving different tasks. This isolates future users from the costly steps of obtaining, storing, manipulating, and processing imagery themselves. The magnitude of the resulting benefits grow with the size of the expanding SIML user community and the scale of global imagery data, which currently increases by more than 80TB/day[15].

## Results
**Achieving accessibility and generalizability with Multi-task Observation using Satellite Imagery & Kitchen Sinks (MOSAIKS).** Our objective is to enable any user with basic resources to predict ground conditions using only satellite imagery and a limited sample of task-specific ground truth data which they possess. Our SIML system, "Multi-task Observation using Satellite Imagery and Kitchen Sinks" (MOSAIKS, see Methods), makes SIML accessible and generalizable by separating the prediction procedure into two independent steps: a fixed "featurization step" which translates satellite imagery into succinct vector representations ($images \rightarrow x$), and a "regression step" which learns task-specific coefficients that map these features to outcomes for a given task ($x \rightarrow y$). For each image, the unsupervised featurizaton step can be centrally executed once, producing one set of outputs that are used to solve many different tasks through repeated application of the regression step by multiple independent users (Fig. 1b). Because the regression step is computationally efficient, MOSAIKS scales nearly costlessly across unlimited users and tasks.

The *accessibility* of our approach stems from the simplicity and computational efficiency of the regression step for potential users, given features which are already computed once and stored centrally. To generate SIML predictions, a user of MOSAIKS (i) queries these tabular data for a vector of $K$ features for each of their $N$ locations of interest; (ii) merges these features $x$ with label data $y$, i.e., the user's independently collected ground truth data; (iii) implements a linear regression of $y$ on $x$ to obtain coefficients $\beta$ – below, we use ridge regression; (iv) uses coefficients $\beta$ and and features $x$ to predict labels $\hat{y}$ in new locations where imagery and features are available but ground truth data are not (Fig.1b).

The *generalizability* of our approach means that a single mathematical summary of satellite imagery ($x$) performs well across many prediction tasks ($y_1, y_2, \ldots$) without any task-specific modification to the procedure. The success of this generalizability relies on how images are encoded as features. We design a featurization function by building on the theoretically grounded machine learning concept of random kitchen sinks[16], which we apply to satellite imagery by constructing random convolutional features (RCFs) (Fig. 1c, Methods). RCFs are suitable for the structure of satellite imagery and have established performance encoding genetic sequences[17], classifying photographs[18], and predicting solar flares[19] (see Supplementary Note 2.3). RCFs capture a flexible measure of similarity between every sub-image across every pair of images without using contextual or task-specific information. The regression step in MOSAIKS then treats these features $x$ as an overcomplete basis for predicting any $y$, which may be a nonlinear function of image elements (see Methods).

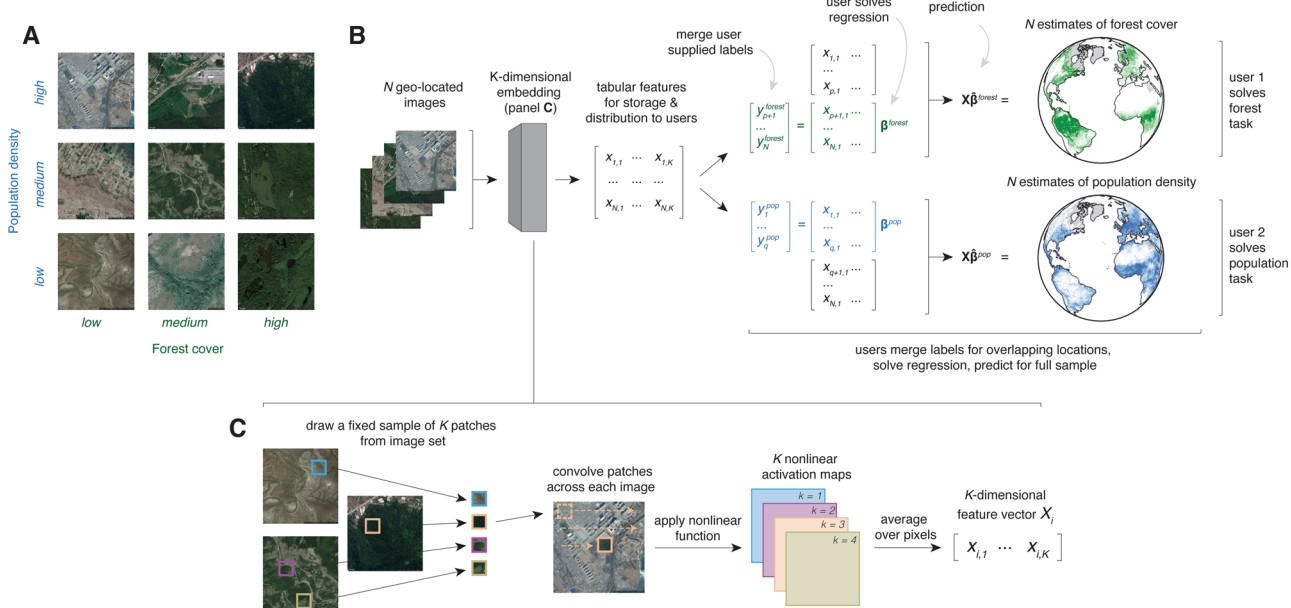

**Fig. 1 A generalizable approach to combining satellite imagery with machine learning (SIML) without users handling images.** MOSAIKS is designed to solve an unlimited number of tasks at planet-scale quickly. After a one-time unsupervised image featurization using random convolutional features, MOSAIKS centrally stores and distributes task-agnostic features to users, each of whom generates predictions in a new context. **a** Satellite imagery is shared across multiple potential tasks. For example, nine images from the US sample are ordered based on population density and forest cover, both of which have distinct identifying features that are observable in each image. **b** Schematic of the MOSAIKS process. *N* images are transformed using random convolutional features into a compressed and highly descriptive *K*-dimensional feature vector before labels are known. Once features are computed, they can be stored in tabular form (matrix **X**) and used for unlimited tasks without recomputation. Users interested in a new task (*s*) merge their own labels ($y^s$) to features for training. Here, user 1 has forest cover labels for locations $p + 1$ to *N* and user 2 has population density labels for locations 1 to *q*. Each user then solves a single linear regression for $\boldsymbol{\beta}^s$. Linear prediction using $\boldsymbol{\beta}^s$ and the full sample of MOSAIKS features **X** then generates SIML estimates for label values at all locations. Generalizability allows different users to solve different tasks using an identical procedure and the same table of features—differing only in the user-supplied label data for training. Each task can be solved by a user on a desktop computer in minutes without users ever manipulating the imagery. **c** Illustration of the one-time unsupervised computation of random convolutional features (Methods and Supplementary Note 2.3). *K* patches are randomly sampled from across the *N* images. Each patch is convolved over each image, generating a nonlinear activation map for each patch. Activation maps are averaged over pixels to generate a single *K*-dimensional feature vector for each image.

In contrast to many recent alternative approaches to SIML, MOSAIKS does not require training or using the output of a deep neural network and encoding images into unsupervised features requires no labels. Nonetheless, MOSAIKS achieves competitive performance at a large computational advantage that grows linearly with the number of SIML users and tasks, due to shared computation and storage. In principle, any unsupervised featurization would enable these computational gains. However, to date, a single set of unsupervised features has neither achieved accuracy competitive with supervised CNN-based approaches across many SIML tasks, nor at the scale that we study. Below, we show that MOSAIKS achieves a practical level of generalization in real-world contexts.

We design a battery of experiments to test whether and under what settings MOSAIKS can provide access to high-performing, computationally efficient, global-scale SIML predictions. Specifically, we (1) demonstrate generalization across tasks, and compare MOSAIKS's performance and cost to existing state-of-the-art SIML models; (2) assess its performance when data are limited and when predicting far from observed labels; (3) scale the analysis to make global predictions and try recreating the results of a national survey; and (4) detail additional properties of MOSAIKS, such as the ability to make predictions at finer resolution than the provided labels.

**Multi-task performance of MOSAIKS in the US**. We first test whether MOSAIKS achieves a practical level of generalization by applying it to a diverse set of pre-selected tasks in the United States (US). While many applications of interest for SIML are in remote and/or data-limited environments where ground truth may be unavailable or inaccurate, systematic evaluation and validation of SIML methods are most reliable in well-observed and data-rich environments[20].

We sample daytime images using the Google Static Maps API from across the continental US ($N = 100,000$), each covering ~1 km × 1 km (256-by-256 pixels) (Supplementary Notes 2.1–2.2). We first implement the featurization step, passing these images through MOSAIKS' feature extraction algorithm to produce $K = 8,192$ features per image (Supplementary Note 2.3). Using only the resulting matrix of features (**X**), we then repeatedly implement the regression step by solving a cross-validated ridge regression for each task and predict forest cover ($R^2 = 0.91$), elevation ($R^2 = 0.68$), population density ($R^2 = 0.72$), nighttime lights ($R^2 = 0.85$), average income ($R^2 = 0.45$), total road length ($R^2 = 0.53$), and average house price ($R^2 = 0.52$) in a holdout test sample (Fig. 2, Supplementary Table 2, Supplementary Notes 2.4–2.6). Computing the feature matrix **X** from imagery took less than 2 hours on a cloud computing node (Amazon EC2 p3.2xlarge instance, Tesla V100 GPU). Subsequently, solving a cross-validated ridge regression for each task took 6.8 min to compute on a local workstation with ten cores (Intel Xeon CPU E5-2630) (Supplementary Note 3.2). These seven outcomes are not strongly correlated with one another (Supplementary Fig. 2) and no attempted tasks in this experiment

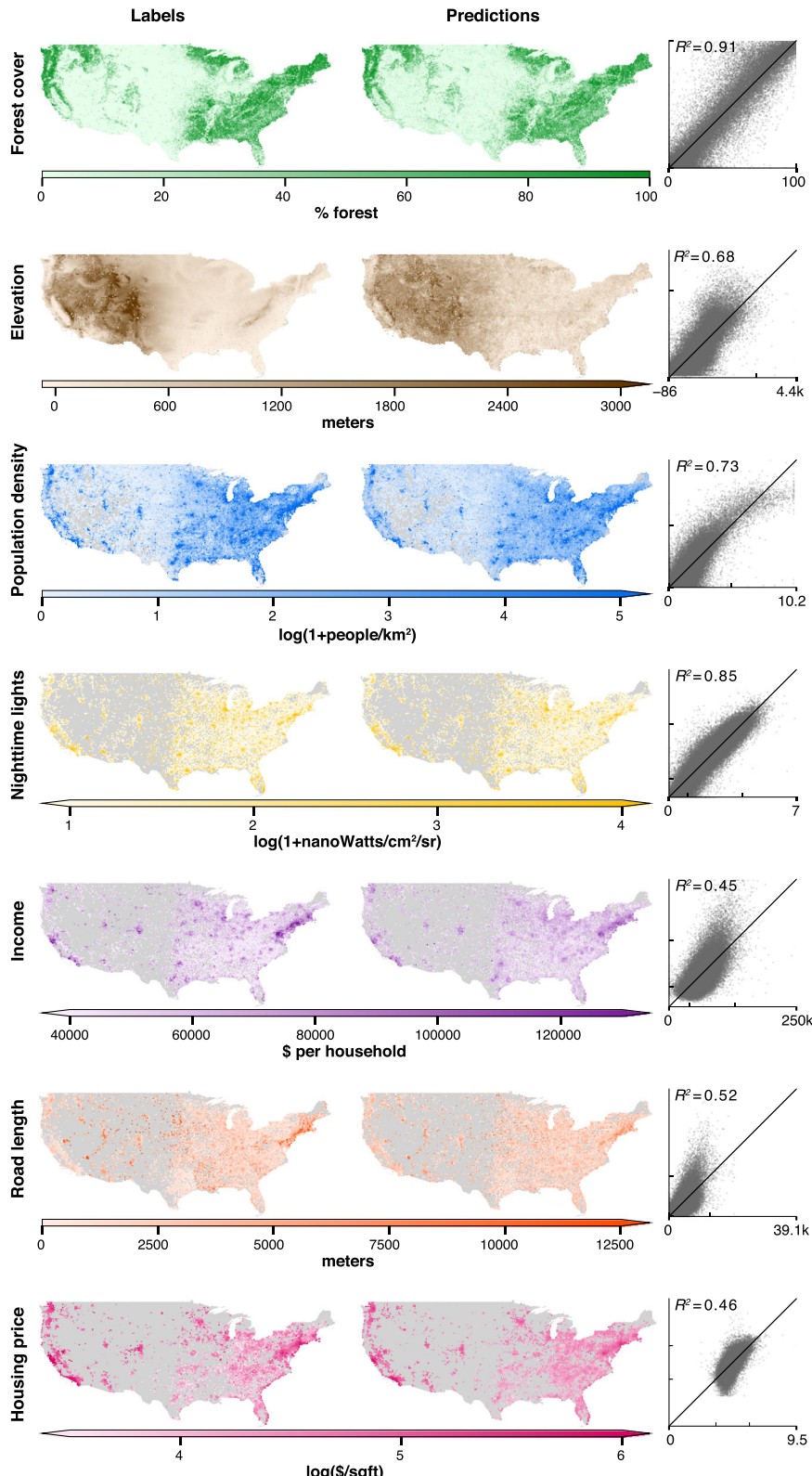

**Fig. 2 1 km × 1 km resolution prediction of many tasks across the continental US using daytime images processed once, before tasks were chosen.**
100,000 daytime images were each converted to 8,192 features and stored. Seven tasks were then selected based on coverage and diversity. Predictions were generated for each task using the same procedure. Left maps: 80,000 observations used for training and validation, aggregated up to 20 km × 20 km cells for display. Right maps: concatenated validation set estimates from 5-fold cross-validation for the same 80,000 grid cells (observations are never used to generate their own prediction), identically aggregated for display. Scatters: Validation set estimates (vertical axis) vs. ground truth (horizontal axis); each point is a ~1 km × 1 km grid cell. Black line is at 45°. Test-set and validation set performance are essentially identical (Supplementary Table 2); validation set values are shown for display purposes only, as there are more observations. The tasks in the top three rows are uniformly sampled across space, the tasks in the bottom four rows are sampled using population weights (Supplementary Note 2.1); grey areas were not sampled in the experiment.

are omitted. These results indicate that MOSAIKS is skillful for a wide range of possible applications without changing the procedure or features and without task-specific expertise. Note that due to the absence of metadata describing the exact time of observation in the Google imagery, as well as task-specific data availability constraints, these performance measures are conditional on a certain degree of unknown temporal mismatch between imagery and task labels (Supplementary Note 1).

**Comparison to state-of-the-art SIML approaches**. We contextualize this performance by comparing MOSAIKS to existing deep-learning based SIML approaches. First, we retrain end-to-end a commonly-used deep convolutional neural network (CNN) architecture[21–23] (ResNet-18) using identical imagery and labels for the seven tasks above. This training took 7.9 hours per task on a cloud computing node (Amazon EC2 p3.xlarge instance, Tesla V100 GPU). We find that MOSAIKS exhibits predictive accuracy competitive with the CNN for all seven tasks (mean $R^2_{CNN} - R^2_{MOSAIKS} = 0.04$; smallest $R^2_{CNN} - R^2_{MOSAIKS} = -0.03$ for housing; largest $R^2_{CNN} - R^2_{MOSAIKS} = 0.12$ for elevation) in addition to being ~250–10,000 × faster to train, depending on whether the regression step is performed on a laptop (2018 Macbook Pro) or on the same cloud computing node used to train the CNN (Fig. 3a, Supplementary Note 3.1 and Supplementary Table 8).

Second, we apply transfer learning[24] using the ResNet-152 CNN pre-trained on natural images to featurize the same satellite images[22,23]. We then apply ridge regression to the CNN-derived features. The speed of this approach is similar to MOSAIKS, but its performance is dramatically lower on all seven tasks (Fig. 3a, Supplementary Note 3.1).

Third, we compare MOSAIKS to an approach from prior studies[4,13,25] where a deep CNN (VGG16[26] pretrained on the ImageNet dataset) is trained end-to-end on night lights and then each task is solved via transfer learning (Supplementary Note 3.1). We apply MOSAIKS to the imagery from Rwanda, Haiti, and Nepal used in ref. [13] to solve all eleven development-oriented tasks they analyze. We find MOSAIKS matches prior performance across tasks in Rwanda and Haiti, and has slightly lower performance (average $\Delta R^2 = 0.08$) on tasks in Nepal (Supplementary Fig. 16). The regression step of this transfer learning approach and MOSAIKS are similarly fast, but the transfer learning approach requires country-specific retraining of the CNN, limiting its accessibility and reducing its generalizability.

Together, these three experiments illustrate that with a single set of task-independent features, MOSAIKS predicts outcomes across a diverse set of tasks, with performance and speed that favorably compare to existing SIML approaches. However, throughout this set of experiments, we find that some sources of variation in labels are not recovered by MOSAIKS. For example, extremely high elevations (>3,000 m) are not reliably distinguished from high elevations (2,400-3,000m) that appear visually similar (Supplementary Fig. 9). Additionally, roughly half the variation in incomes and housing prices is unresolved, presumably because they depend on factors not observable from orbit, such as tax policies or school districts (Fig. 2).

These experiments additionally reveal that patterns of predictability across tasks are strikingly similar in MOSAIKS and in alternative SIML approaches (Supplementary Figs. 16 and 17). Together, these findings are consistent with the hypothesis that there exists some performance ceiling for each task, due to some factors not being observable from satellite imagery. To investigate this further, we develop a hybrid model in which the 512 features produced by the last layer of the ResNet-18 CNN are concatenated with the 8,192 MOSAIKS features and included together in a ridge regression. Performance improvements above

either MOSAIKS or the CNN are small (≤0.01$R^2$) for most tasks, although there is a notable performance boost for the two tasks where both models achieve the lowest accuracy ($R^2_{hybrid} - R^2_{CNN} = 0.04$ for income; $R^2_{hybrid} - R^2_{MOSAIKS} = 0.05$ for housing price; Supplementary Table 7). These results suggest that for some tasks, combining MOSAIKS with alternative SIML models can enhance predictive accuracy.

**Evaluations of model sensitivity**. There is growing recognition that understanding the accuracy, precision, and limits of SIML predictions is important, since consequential decisions increasingly depend on these outputs, such as which households should receive financial assistance[20,27]. However, historically, the high costs of training deep-learning models have generally prevented the stress-testing and bench-marking that would ensure accuracy and constrain uncertainty. To characterize the performance of MOSAIKS, we test its sensitivity to the number of features ($K$) and training observations ($N$), as well as the extent of spatial extrapolation.

Unlike some featurization methods, these is no known measure of importance for individual features in MOSAIKS, so the computational complexity of the regression step can be manipulated by simply including more or fewer features. Repeatedly re-solving the linear regression step in MOSAIKS with a varied number of features indicates that increasing $K$ above 1,000 features provides minor predictive gains (Fig. 3b). A majority of the observable signal in the baseline experiment using $K = 8,192$ is recovered using $K = 200$ (min 55% for income, max 89% for nighttime lights), reducing each 65,536-pixel tri-band image to just 200 features (~250 × data compression). Similarly, re-solving MOSAIKS predictions with a different number of training observations demonstrates that models trained with fewer samples may still exhibit high accuracy (Fig. 3b). A majority of the available signal is recovered for many outcomes using only $N = 500$ (55% for road length to 87% for forest cover), with the exception of income (28%) and housing price (26%) tasks, which require larger samples. Together, these experiments suggest that users with computational, data acquisition, or data storage constraints can easily tailor MOSAIKS to match available resources and can reliably estimate the performance impact of these alterations (Supplementary Note 2.7).

To systematically evaluate the ability of MOSAIKS to make accurate predictions in large contiguous areas where labels are not available, we conduct a spatial cross-validation experiment by partitioning the US into a checkerboard pattern (Fig. 3c), training on the black squares and testing on the white squares (Supplementary Note 2.8). Increasing the width of squares ($\delta$) in the checkerboard increases the average distances between train and test observations, simulating increasingly large spatial extrapolations. We find that for three of seven tasks (forest cover, population density, and nighttime lights), performance declines minimally regardless of distance (maximum $R^2$ decline of 10% at $\delta = 16°$ for population density). For income, road length, and housing price, performance falls moderately at small degrees of spatial extrapolation (19%, 33%, and 35% decline at $\delta = 4°$, respectively), but largely stabilizes thereafter. Note that the poor performance of road length predictions is possibly due to missing labels and data quality (Supplementary Note 1.1 and Supplementary Fig. 1). Finally, elevation exhibits steady decline with increasing distances between training and testing data (49% decline at $\delta = 16°$).

To contextualize this performance, we compare MOSAIKS to spatial interpolation of observations, a widely used approach to fill in regions of missing data (Supplementary Note 2.8). Using the same samples, MOSAIKS substantially outperforms spatial

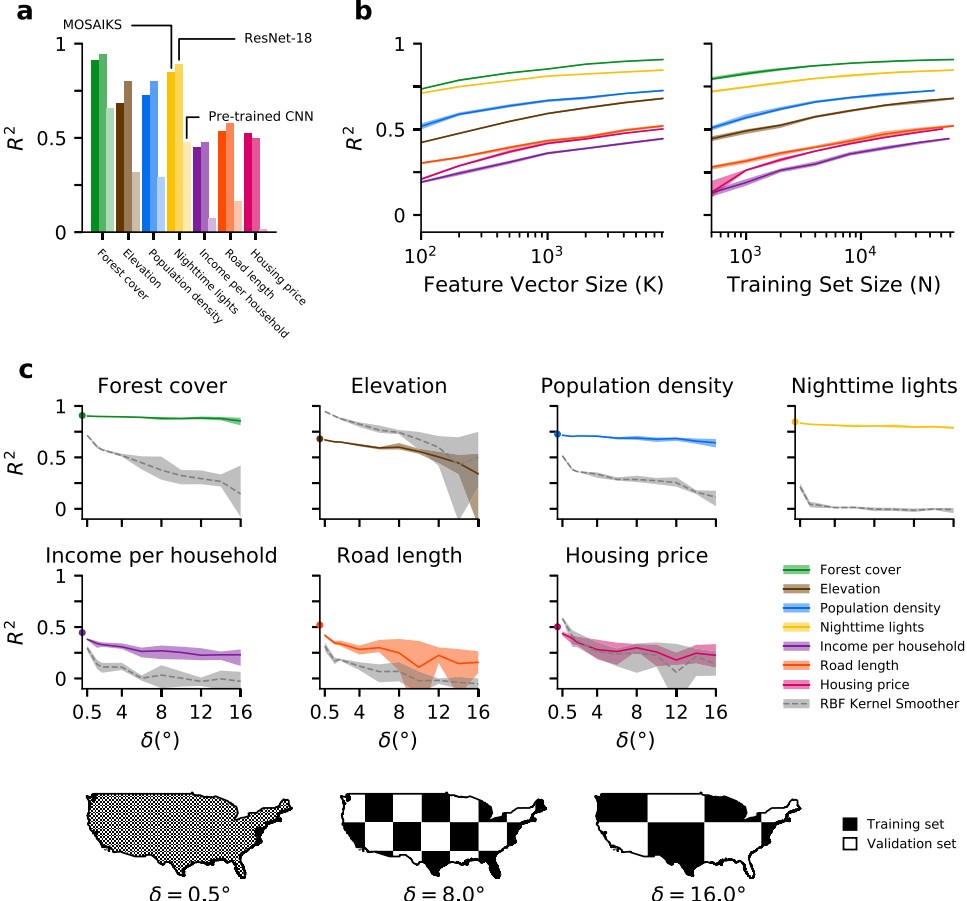

**Fig. 3 Prediction accuracy relative to a convolutional neural network and transfer learning, using smaller K and N, and over large contiguous regions with no ground truth data. a** Task-specific MOSAIKS test-set performance (dark bars) in contrast to: an 18-layer variant of the ResNet Architecture (ResNet-18) trained end-to-end for each task (middle bars); and transfer learning based on an unsupervised featurization using the last hidden layer of a 152-layer ResNet variant pre-trained on natural imagery and applied using ridge regression (lightest bars). See Supplementary Note 3.1 for details. **b** Validation set $R^2$ performance for all seven tasks while varying the number of random convolutional features $K$ and holding $N = 64,000$ (left) and while varying $N$ and holding $K = 8,192$ (right). Shaded bands indicate the range of predictive skill across five folds. Lines indicate average accuracy across folds. **c** Evaluation of performance over regions of increasing size that that are excluded from training sample. Data are split using a checkerboard partition, where the width and height of each square is $\delta$ (measured in degrees). Example partitions with $\delta = 0.5°, 8°, 16°$ are shown in maps. For a given $\delta$, training occurs using data sampled from black squares and performance is evaluated in white squares. Plots show colored lines representing average performance of MOSAIKS in the US across $\delta$ values for each task. Benchmark performance from Fig. 2 are indicated as circles at $\delta = 0$. Grey dashed lines indicate corresponding performance using only spatial interpolation with an optimized radial basis function (RBF) kernel instead of MOSAIKS (Supplementary Note 2.8). To moderate the influence of the exact placement of square edges, training and test sets are resampled four times for each $\delta$ with the checkerboard position re-initialized using offset vertices (see Supplementary Note 2.8 and Supplementary Fig. 10). The ranges of performance are plotted as colored or grey bands.

interpolation (Fig. 3c, grey dashed lines) across all tasks except for elevation, where interpolation performs almost perfectly over small ranges ($\delta = 0.5°$: $R^2 = 0.95$), and housing price, where interpolation slightly outperforms MOSAIKS at small ranges. For both, interpolation performance converges to that of MOSAIKS over larger distances. Thus, in addition to generalizing across tasks, MOSAIKS generalizes out-of-sample across space, outperforming spatial interpolation of ground truth in five of seven tasks.

The above sensitivity tests are enabled by the speed and simplicity of training MOSAIKS. These computational gains also enable quantification of uncertainty in model performance within each diagnostic test. As demonstrated by the shaded bands in Figs. 3b–c, uncertainty in MOSAIKS performance due to variation in splits of training-validation data remains modest under most conditions.

**Applying MOSAIKS at scale**. Having evaluated MOSAIKS systematically in the data-rich US, we test its performance at planetary scale and its ability to recreate results from a national survey.

We test the ability of MOSAIKS to scale globally using the four tasks for which global labels are readily available. Using a random sub-sample of global land locations (training and validation: $N = 338,781$, test: $N = 84,692$; Supplementary Note 2.10), we construct the first planet-scale, multi-task estimates using a single set of label-independent features ($K = 2,048$, Fig. 4a), predicting the distribution of forest cover ($R^2 = 0.85$), elevation ($R^2 = 0.45$), population density ($R^2 = 0.62$), and nighttime lights ($R^2 = 0.49$). Note that inconsistent image and label quality across the globe are likely partially responsible for lowering performance relative to the US-only experiments above (Supplementary Note 2.10).

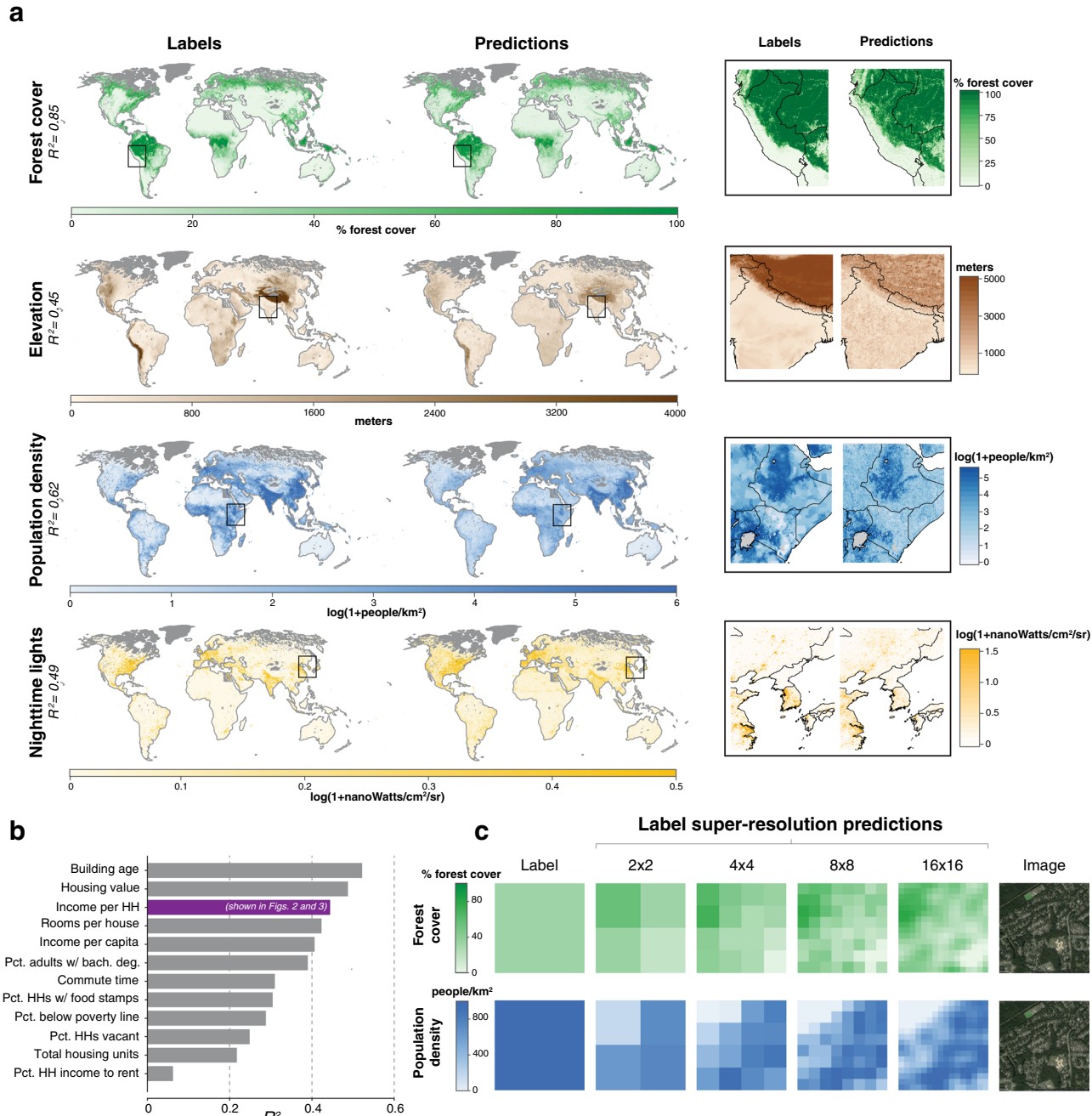

**Fig. 4 A single featurization of imagery predicts multiple variables at planet-scale, predicts results from a national survey, and achieves label super-resolution. a** Training data (left maps) and predictions using a single featurization of daytime imagery (right maps). Insets (far right) marked by black squares in global maps. Training sample is a uniform random sampling of 1,000,000 land grid cells, 498,063 for which imagery were available and could be matched to task labels. Out-of-sample predictions are constructed using five-fold cross-validation. For display purposes only, maps depict ~50 km × 50km average values (ground truth and predictions at ~1 km × 1 km). **b** Test-set performance in the US shown for 12 variables from the 2015 American Community Survey (ACS) conducted by the US Census Bureau[31]. Income per household (HH) (in purple) is also shown in Figs. 2 and 3, and was selected as an outcome for the analysis in those figures before this ACS experiment was run. **c** Both labels and features in MOSAIKS are linear combinations of sub-image ground-level conditions, allowing optimized regression weights to be applied to imagery of any spatial extent (Supplementary Note 2.9). MOSAIKS thus achieves label super-resolution by generating skillful estimates at spatial resolutions finer than the labels used for training. Shown are example label super-resolution estimates at 2 × 2, 4 × 4, 8 × 8, and 16 × 16, along with the original 1 × 1 label resolution (See Supplementary Fig. 12 for additional examples). Systematic evaluation of within-image $R^2$ across the entire sample is reported in Supplementary Note 2.9 for the forest cover task.

It has been widely suggested that SIML could be used by resource-constrained governments to reduce the cost of surveying their citizens[4,13,28–30]. To demonstrate MOSAIKS's performance in this theoretical use-case, we simulate a field test with the goal of recreating results from an existing nationally representative survey. Using the pre-computed features from the first US experiment above, we generate predictions for 12 pre-selected questions in the 2015 American Community Survey (ACS) conducted by the US Census Bureau[31]. We obtain $R^2$ values ranging from 0.06 (percent household income spent on rent, an

outlier) to 0.52 (building age), with an average $R^2$ of 0.34 across 12 tasks (Fig. 4b). Compared to a baseline of no ground survey, or a costly survey extension, these results suggest that MOSAIKS predictions could provide useful information to a decision-maker for almost all tasks at low cost; noting that, in contrast, the ACS costs >$200 million to deploy annually[32]. However, some variables (e.g., percent household income spent on rent) may continue to be retrievable only via ground survey.

**Methodological extensions**. The design of MOSAIKS naturally provides two additional useful properties: suitability to fusing features with data from other sensors, and the ability to attribute image-scale predictions to sub-image level regions.

Available satellites exhibit a diversity of properties (e.g., wavelength, timing of sampling) that can be used to improve SIML predictions[33]. While most SIML approaches, including the above analysis, use a single sensor, the design of MOSAIKS allows seamless integration of data from additional satellites because the regression step is linear in the features. To demonstrate this, we include nighttime lights as a second data source in the analysis of survey data from Rwanda, Haiti, and Nepal discussed above (Supplementary Note 3.1). The approach mirrors that of the hybrid MOSAIKS-ResNet18 model discussed previously in that features extracted from the nighttime lights data are simply concatenated with those from MOSAIKS prior to the regression step. In all 36 tasks, predictions either improved or were unchanged when nighttime imagery was added to daytime imagery in the model (average $\Delta R^2 = 0.03$). This approach naturally optimizes how data from all sensors are used without requiring that users possess expertise on each technology.

Many use cases would benefit from SIML predictions at finer resolution than is available in training data[33,34]. Here we show that MOSAIKS can estimate the relative contribution of sub-regions within an image to overall image-level labels, even though only aggregated image-level labels are used in training (See Fig. 4c and Supplementary Fig. 12). Such label super-resolution prediction follows from the functional form of the featurization and linear regression steps in MOSAIKS, allowing it to be analytically derived for labels that represent nearly linear combinations of ground-level conditions (Supplementary Note 2.9 and Supplementary Fig. 11). We numerically assess label super-resolution predictions of MOSAIKS for the forest cover task, since raw label data are available at much finer resolution than our image labels. Provided only a single label per image, MOSAIKS recovers substantial within-image signal when predicting forest cover in 4 to 1024 sub-labels per label (within-image $R^2 = 0.54$–0.32, see Supplementary Fig. 13 for a plot of performance against number of sub-labels and Supplementary Note 2.9 for m_ethodological details).

## Discussion

We develop a new approach to SIML that achieves practical generalization across tasks while exhibiting performance that is competitive with deep-learning models optimized for a single task. Crucial to planet-scale analyses, MOSAIKS requires orders of magnitude less computation time to solve a new task than CNN-based approaches and it allows 1km-by-1km image data to be compressed ~6–500 times before storage/transmission (see Methods). Such compression is a deterministic operation that could theoretically be implemented in satellite hardware. We hope these computational gains, paired with the relative simplicity of using MOSAIKS, will democratize access to global-scale SIML technology and accelerate its application to solving pressing global challenges. We hypothesize that there exist hundreds of

variables observable from orbit whose application could improve human well-being if measurements were made accessible.

While we have shown that in many cases MOSAIKS is a faster and simpler alternative to existing deep-learning methods, there remain contexts in which custom-designed SIML pipelines will continue to play a key role in research and decision-making, such as where resources are plentiful and performance is paramount. Existing ground-based surveys will also remain important. In both cases we expect MOSAIKS can complement these systems, especially in resource-constrained settings. For example, MOSAIKS can provide fast assessments to guide slower SIML systems or extend the range and resolution of ground-based surveys.

As real-world policy actions increasingly depend on SIML predictions, it is crucial to understand the accuracy, precision and sensitivity of these measurements. The low cost and high speed of retraining MOSAIKS enables unprecedented stress tests that can support robust SIML-based decision systems. Here, we tested the sensitivity of MOSAIKS to model parameters, number of training points, and degree of spatial extrapolation, and expect that many more tests can be developed and implemented to analyze model performance and prediction accuracies in context. To aid systematic bench-marking and comparison of SIML architectures, the labels and features used in this study are made publicly available; to our knowledge this represents the largest multi-label benchmark dataset for SIML regression tasks. The high performance of RCF, a relatively simple featurization, suggests that developing and bench-marking other unsupervised SIML methods across tasks at scale may be a rich area for future research.

By distilling SIML to a pipeline with simple and mathematically interpretable components, MOSAIKS facilitates development of methodologies for additional SIML use cases and enhanced performance. For example, the ability of MOSAIKS to achieve label super-resolution is easily derived analytically (Supplementary Note 2.9). Furthermore, while we have focused here on tri-band daytime imagery, we showed that MOSAIKS can seamlessly integrate data from multiple sensors through simple concatenation, extracting useful information from each source to maximize performance. We conjecture that integrating new diverse data, from both satellite and non-satellite sources, may substantially increase the predictive accuracy of MOSAIKS for tasks not entirely resolved by daytime imagery alone; such integration using deep-learning models is an active area of research[35].

We hope that MOSAIKS lays the foundation for the future development of an accessible and democratized system of global information sharing, where, over time, imagery from all available global sensors is continuously encoded as features and appended to a single table of data, which is distributed and used planet-wide. As a step in this direction, we make a global cross-section of features publicly available. Such a unified global system may enhance our collective ability to observe and understand the world, a necessary condition for tackling pressing global challenges.

## Methods

**Overview**. Here we provide additional information on our implementation of MOSAIKS and experimental procedures, as well as a description of the theoretical foundation underlying MOSAIKS. Full details on the methodology behind MOSAIKS can be found throughout Supplementary Note 2.

**Implementation of MOSAIKS**. We begin with a set of images $\{\mathbf{I}_\ell\}_{\ell=1}^N$, each of which is centered at locations indexed by $\ell = \{1, \dots, N\}$. MOSAIKS generates task-agnostic feature vectors $\mathbf{x}(\mathbf{I}_\ell)$ for each satellite image $\mathbf{I}_\ell$ by convolving an $M \times M \times S$ "patch", $\mathbf{P}_k$, across the entire image. $M$ is the width and height of the patch in units of pixels and $S$ is number of spectral bands. In each step of the convolution, the inner product of the patch and an $M \times M \times S$ sub-image region is taken, and a ReLU activation function with bias $b_k = 1$ is applied. Each patch is a randomly

sampled sub-image from the set of training images $\{\mathbf{I}_\ell\}_{\ell=1}^N$ (Supplementary Fig. 5). In our main analysis, we use patches of width and height $M = 3$ (Supplementary Fig. 6) and $S = 3$ bands (red, green, and blue). To create a single summary metric for the image-patch pair, these inner product values are then averaged across the entire image, generating the $k$th feature $\mathbf{x}_k(\mathbf{I}_\ell)$, derived from patch $\mathbf{P}_k$. The dimension of the resulting feature space is equal to $K$, the number of patches used, and in all of our main analyses we employ $K = 8{,}192$ (i.e., $2^{13}$). Both images and patches are whitened according to a standard image preprocessing procedure before convolution (Supplementary Note 2.3).

In practice, this one-time featurization can be centrally computed and then features $\mathbf{x}_k(\mathbf{I}_\ell)$ distributed to users in tabular form. A user need only (i) obtain and link the subset of these features that match spatially with their own labels and then (ii) solve linear regressions of the labels on the features to learn nonlinear mappings from the original image pixel values to the labels (the nonlinearity of the mapping between image pixels and labels stems from the nonlinearity of the ReLU activation function). We show strong performance across seven different tasks using ridge regression to train the relationship between labels $y_\ell$ and features $\mathbf{x}_k(\mathbf{I}_\ell)$ in this second step, although future work may demonstrate that other fitting procedures yield similar or better results for particular tasks.

Implementation of this one-time unsupervised featurization takes about the same time to compute as a single forward pass of a CNN. With $K = 8{,}912$ features, featurization results in a roughly 6 to 1 compression of stored and transmitted imagery data in the cases we study. Notably, storage and computational cost can be traded off with performance by using more or fewer features from each image (Fig. 3b). Since features are random, there is no natural value for $K$ that is specifically preferable.

**Task selection and data**. Tasks were selected based on diversity and data availability, with the goal of evaluating the generalizability of MOSAIKS (Supplementary 1.1). Results for all tasks evaluated are reported in the paper. We align image and label data by projecting imagery and label information onto a ~1 km × 1 km grid, which was designed to ensure zero spatial overlap between observations (Supplementary Notes 2.1 and 2.2).

Images are obtained from the Google Static Maps API (Supplementary Note 1.2)[36], and labels for the seven tasks are obtained from refs. [2,31,37–41]. Details on data are described in Supplementary Table 1 and Supplementary Note 1.

**US experiments**. From this grid we sample 20,000 hold-out test cells and 80,000 training and validation cells from within the continental US (Supplementary Note 2.4). To span meaningful variation in all seven tasks, we generate two of these 100,000-sample data sets according to different sampling methods. First, we sample uniformly at random across space for the forest cover, elevation, and population density, tasks which exhibit rich variation across the US. Second, we sample via a population-weighted scheme for nighttime lights, income, road length, and housing price, tasks for which meaningful variation lies within populated areas of the US. Some sample sizes are slightly reduced due to missing label data ($N = 91{,}377$ for income, 80,420 for housing price, and 67,968 for population density). We model labels whose distribution is approximately log-normal using a log transformation (Supplementary Note 2.5 and Supplementary Table 3).

Because fitting a linear model is computationally cheap, relative to many other SIML approaches, it is feasible to conduct numerous sensitivity tests of predictive skill. We present cross-validation results from a random sample, while also systematically evaluating the behavior of the model with respect to: (a) geographic distance between training and testing samples, i.e., spatial cross-validation, (b) the dimension $K$ of the feature space, and (c) the size $N$ of the training set (Fig. 3, Supplementary Notes 2.7 and 2.8). We represent uncertainty in each sensitivity test by showing variance in predictive performance across different training and validation sets. We also benchmark model performance and computational expense against an 18-layer variant of the ResNet Architecture, a common deep network architecture that has been used in satellite-based learning tasks[42], trained end-to-end for each task and a transfer learning approach[24] utilizing an unsupervised featurization based on the last hidden layer of a 152-layer ResNet variant pre-trained on natural imagery and applied using ridge regression (Supplementary Notes 3.1 and 3.2).

**Global experiment**. To demonstrate performance at scale, we apply the same approach used within the data-rich US context to global imagery and labels. We employ a target sample of $N = 1{,}000{,}000$, which drops to a realized sample of $N = 423{,}476$ due to missing imagery and label data outside the US (Fig. 4). We generate predictions for all tasks with labels that are available globally (forest cover, elevation, population density, and nighttime lights) (Supplementary Note 2.10).

**Label super-resolution experiment**. Predictions at label super-resolution (i.e., higher resolution than that of the labels used to train the model), shown in Fig. 4c, are generated for forest cover and population density by multiplying the trained ridge regression weights by the un-pooled feature values for each sub-image and applying a Gaussian filter to smooth the resulting predictions (Supplementary Note 2.9). Additional examples of label super-resolution performance are shown in Supplementary Fig. 12. We quantitatively assess label super-resolution

performance (Supplementary Fig. 13) using forest cover, as raw forest cover data are available at substantially finer resolution than our common ~1 km × 1 km grid. Performance is evaluated by computing the fraction of variance ($R^2$) within each image that is captured by MOSAIKS, across the entire sample.

**Theoretical foundations**. MOSAIKS is motivated by the goal of enabling generalizable and skillful SIML predictions. It achieves this by embedding images in a basis that is both descriptive (i.e., models trained using this single basis achieve high skill across diverse labels) and efficient (i.e., such skill is achieved using a relatively low-dimensional basis). The approach for this embedding relies on the theory of random kitchen sinks[16], a method for feature generation that enables the linear approximation of arbitrary well-behaved functions. This is akin to the use of polynomial features or discrete Fourier transforms for function approximation generally, such as functions of one dimension. When users apply these features in linear regression, they identify linear weightings of these basis vectors important for predicting a specific set of labels. With inputs of high dimension, such as the satellite images we consider, it has been shown experimentally[17–19] and theoretically[43] that a randomly selected subspace of the basis often performs as well as the entire basis for prediction problems.

**Convolutional random kitchen sinks**. Random kitchen sinks approximate arbitrary functions by creating a finite series of features generated by passing the input variables $z$ through a set of $K$ nonlinear functions $g(z; \Theta_k)$, each parameterized by draws of a random vector $\Theta$. The realized vectors $\Theta_k$ are drawn independently from a pre-specified distributions for each of $k = 1\ldots K$ features. Given an expressive enough function $g$ and infinite $K$, such a featurization would be a universal function approximator[43]. In our case, such a function $g$ would encode interactions between all subsets of pixels in an image. Unfortunately, for an image of size $256 \times 256 \times 3$, there are $2^{256 \times 256 \times 3}$ such subsets. Therefore, the fully-expressive approach is inefficient in generating predictive skill with reasonably concise $K$ because each feature encodes more pixel interactions than are empirically useful.

To adapt random kitchen sinks for satellite imagery, we use convolutional random features, making the simplifying assumption that most information contained within satellite imagery is represented in local image structure. Random convolutional features have been shown to provide good predictive performance across a variety of tasks from predicting DNA binding sites[17] and solar flares[19] to clustering photographs[18] (kitchen sinks have also been used in a non-convolutional approach to classify individual pixels of hyper-spectral satellite data[44]). Applied to satellite images, random convolutional features reduce the number of effective parameters in the function by considering only local spatial relationships between pixels. This results in a highly expressive, yet computationally tractable, model for prediction.

Specifically, we create each $\Theta_k$ by extracting a small sub-image patch from a randomly selected image within our image set, as described above. These patches are selected independently, and in advance, of any of the label data. The convolution of each patch across the satellite image being featurized captures information from the entire $\mathbb{R}^{256 \times 256 \times 3}$ image using only $3 * M^2$ free parameters for each $k$. Creating and subsequently averaging over the activation map (after a ReLU nonlinearity) defines our instantiation of the kitchen sinks function $g(z; \Theta_k)$ as $g(\mathbf{I}_\ell; \mathbf{P}_k, b_k) = \mathbf{x}_k(\mathbf{I}_\ell)$, where $b_k$ is a scalar bias term. Our choice of this functional form is guided by both the structural properties of satellite imagery and the nature of common SIML prediction tasks, and it is validated by the performance demonstrated across tasks.

**Relevant structural properties of satellite imagery and SIML tasks**. Three particular properties provide the motivation for our choice of a convolution and average-pool mapping to define $g$.

First, we hypothesize that convolutions of small patches will be sufficient to capture nearly all of the relevant spatial information encoded in images because objects of interest (e.g., a car or a tree) tend to be contained in a small sub-region of the image. This is particularly true in satellite imagery, which has a much lower spatial resolution that most natural imagery (Supplementary Fig. 6).

Second, we expect a single layer of convolutions to perform well because satellite images are taken from a constant perspective (from above the subject) at a constant distance and are (often) orthorectified to remove the effects of image perspective and terrain. Together, these characteristics mean that a given object will tend to appear the same when captured in different images. This allows for MOSAIKS's relatively simple, translation invariant featurization scheme to achieve high performance, and avoids the need for more complex architectures designed to provide robustness to variation in object size and orientation.

Third, we average-pool the convolution outputs because most labels for the types of problems we study can be approximately decomposed into a sum of sub-image characteristics. For example, forest cover is measured by the percent of total image area covered in forest, which can equivalently be measured by averaging the percent forest cover across sub-regions of the image. Labels that are strictly averages, totals, or counts of sub-image values (such as forest cover, road length, population density, elevation, and night lights) will all exhibit this decomposition. While this is not strictly true of all SIML tasks, for example income and average

housing price, we demonstrate that MOSAIKS still recovers strong predictive skill on these tasks. This suggests that some components of the observed variance in these labels may still be decomposable in this way, likely because they are well-approximated by functions of sums of observable objects.

**Additional interpretations**. The full MOSAIKS platform, encompassing both featurization and linear prediction, bears similarity to a few related approaches. Namely, it can be interpreted as a computationally feasible approximation of kernel ridge regression for a fully convolutional kernel or, alternatively, as a two-layer CNN with an incredibly wide hidden layer generated with untrained filters. A discussion of these interpretations and how they can help to understand MOSAIKS's predictive skill can be found in Supplementary Note 2.3.

## Data availability

Code, data, a configured computing environment, and free cloud computing for this analysis is provided via Code Ocean and is available at https://doi.org/10.24433/CO.8021636.v2. All data used in this analysis are from publicly available sources other than the house price data. House price data are provided by Zillow through the Zillow Transaction and Assessment Dataset (ZTRAX) and are used under license for the current study. More information on accessing the data can be found at http://www.zillow.com/ztrax. The results and opinions are those of the author(s) and do not reflect the position of Zillow Group. The house price dataset we release publicly is a subset of the pre-processed and aggregated data used in the analysis, where grid cells containing <30 observations of recent property sales are removed to preserve privacy. While the rest of the data that support the findings of this study are publicly available, the re-dissemination of some of these data is restricted. Thus, we are not able to host all data used in the study within our Code Ocean capsule. For example, both imagery and some label data must be downloaded directly from the original providers. Whenever this is the case, we provide download instructions in the code repository's Readme. In addition to the data directly supporting this study, we provide MOSAIKS features for a gridded cross-section of the globe. This service and any related work will be accessible via http://www.globalpolicy.science/mosaiks.

## Code availability

The code used in this analysis is provided in the github repository available at https://github.com/Global-Policy-Lab/mosaiks-paper and additionally at https://doi.org/10.24433/CO.8021636.v2. The latter is part of the Code Ocean capsule, additionally containing data and computing environment (see Data Availability). On GitHub, release "v1.0" corresponds to the state of the codebase at the time of publication. See the repository's Readme for more detailed information.

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

## Acknowledgements

We thank Patrick Baylis, Joshua Blumenstock, Jennifer Burney, Hannah Druckenmiller, Jonathan Kadish, Alyssa Morrow, James Rising, Geoffrey Schiebinger, Adam Storeygard and participants in seminars at UC Berkeley, University of Chicago, Harvard, American Geophysical Union, the World Bank, the United Nations Development Program & Environment Program, Planet Inc., The Potsdam Institute for Climate Impact Research, the National Bureau of Economic Research, and The Workshop in Environmental Economics and Data Science for helpful comments and suggestions. We acknowledge funding from the NSF Graduate Research Fellowship Program (Grant DGE 1752814), the US Environmental Protection Agency Science To Achieve Results Fellowship Program (Grant FP91780401), the NSF Research Traineeship Program Data Science for the 21st Century, the Harvard Center for the Environment, the Harvard Data Science Initiative, the Sloan Foundation, and a gift from the Rhodium Group. The authors declare no conflicts of interest.

## Author contributions

E.R., J.P., T.C., I.B., V.S., B.R. and S.H. formulated the research idea and designed the overall analysis structure. V.S. collected imagery data and designed and implemented the featurization. J.P., T.C., I.B. and M.I. collected label data. E.R., J.P., T.C., I.B., V.S. and M.I. developed and carried out experimental procedures. E.R., J.P., T.C., I.B., V.S., M.I., B.R. and S.H. analyzed and interpreted the output of the experiments. E.R., J.P., T.C., I.B. and S.H. wrote the paper with contributions from V.S. and B.R.

## Competing interests

The authors declare no competing interests.
