## [Peer Review File · Nature Communications]

REVIEWER COMMENTS

Reviewer #1 (Remarks to the Author):

This paper demonstrates that a universal set of features extracted from satellite images provides predictive accuracy comparable to state-of-the-art methods across a variety of policy-relevant tasks, while requiring orders of magnitude less computing resources. In recent years, satellite imagery has become a key data source used to complement traditional surveys in data-scarce environments, allowing to predict with high accuracy cross-sectional variations in policy-relevant outcomes, from population density to the level of urbanization. However, a key roadblock in the wide-scale adoption of indicators derived from satellite images is that the barrier to entry is high, both in terms of computing resources as well as in well-trained computer scientists capable of analyzing them, especially in low-income countries where they arguably provide the most value. In this context, the method proposed by the authors could allow a wide variety of social scientists to derive insights from these images without having to directly analyse them. Beyond the methodological innovation described in the paper, the authors will make their features publicly available, which could have a high impact among development practitioners and for future research in this space.

I have 2 practical suggestions which could strengthen the results of the paper:

- 1- The authors argue that the MOZAIKS features only need to be computed and stored once. However, the physical characteristics measured in the images change over time, which could deteriorate the quality of the predictions produced by features computed using images from the past. I would encourage the authors to show the time-stability of the MOZAIKS features.
- 2- While the authors provided an extensive comparison of their approach with other SIML approaches, it would be interesting to show whether using the MOZAIKS features as additional features could help improve the predictive accuracy of the other SIML models they analysed.

Reviewer #2 (Remarks to the Author):

The major novelty of this work, from a methodological point of view, lays in the demonstration that a "universal" unsupervised feature extraction process can be applied to satellite image observations, which can then be utilized, in a cost-effective way, to estimate a diverse set of geo-bio physical parameters. Using as input satellite observations from Google Earth, the paper proposed a very detailed and extensive experimental analysis of the behavior of the proposed approach, including the ability to make predictions to new locations where imagery is available but ground truth data may not be. The paper is very well written, the motivation of this work is founded, the experimental analysis is very thorough, and the conclusions are justified.

An important issue regarding the evaluation protocol however is the known limitation when considering google map images, in that neighboring spatial locations (even within the 256x256 region considered here) in a single input image, could have been acquired at distinct, and in general unknown, time instances. On the other hand, "ground" labels are obtained during specific periods in time. How does the proposed approach account for the fact that the input (satellite imagery) and the predictions can be characterized by a varying degree of temporal affinity? Results however demonstrate that the performance, even under the image acquisition uncertainty, is impressive. If it is possible to obtain the time of imaging from Google images, then an analysis of the impact in terms of classification performance would be very convincing. Comparison with alternative paradigms, e.g., pre-training CNNs on natural images, or using models from one task for another, presented in Section 4 of the supplementary material, could benefit from concrete performance metrics (potentially formatted in tables), in addition to figures likes S16 or S17, which makes it difficult quantify the absolute performance gains/losses of the proposed approach.

Another aspect of the analysis which could also benefit from more concert result involved the process of "label super-resolution". In this case, a plot showing the error (e.g. R2) with respect to the super-resolution factor would be ideal.

Minor: in supplementary material, on section "label super-resolution experiment" you refer to Figure 4B, but

I think you mean Figure 4C.

Reviewer #3 (Remarks to the Author):

This paper presents a generalizable and accessible machine learning approach for diverse prediction tasks with satellite images, e.g., forest cover, house price, road length, etc. Authors mentioned that the proposed method shows the competitive performance with deep neural networks and even better in magnitude lower computational cost, scales globally, delivers label super-resolution predictions, and facilitates characterizations of uncertainty. However, the reviewer has the specific comments as follows.

- 1) In the introduction part, a detailed description of various satellite images is missing. For example, analyzing and discussing the advantages and disadvantages of different data sources are needed. This can help readers understand your points or motivations, e.g., why the proposed method is effective to be designed for satellite images and further for different applications, thereby understanding our environment.
- 2) Authors claimed the proposed method is superior to the deep learning-based methods. However, there is a lack of theoretical proof and also experimental results are insufficient since only a few compared methods are investigated, which is relatively hard to support the opinion.
- 3) The reviewer is wondering how about the efficiency (running time and computational cost)? since the authors mentioned the proposed method is more faster and efficient.
- 4) In recent years, lots of advanced deep learning methods and benchmark datasets have been developed. The reviewer suggests discussing and analyzing these advanced and latest methods and also adding more experimental results to show the superiority of the proposed method, such as "More diverse means better: Multimodal deep learning meets remote-sensing imagery classification. *IEEE Transactions on Geoscience and Remote Sensing*, 2020", "Deep learning for remote sensing image classification: A survey. *Wiley Interdisciplinary Reviews: Data Mining and Knowledge Discovery*, 2018, 8(6), e1264.", "Comprehensive survey of deep learning in remote sensing: theories, tools, and challenges for the community. *Journal of Applied Remote Sensing*, 2017, 11(4), 042609.", and "A survey on deep learning-driven remote sensing image scene understanding: Scene classification, scene retrieval and scene-guided object detection. *Applied Sciences*, 2019, 9(10), 2110."

March 2021

POINT-BY-POINT REVIEWER REPLY

Reviewer #1

(original comments in gray and italicized, replies in black and indented,
new text added to manuscript in bold)

This paper demonstrates that a universal set of features extracted from satellite images provides predictive accuracy comparable to state-of-the-art methods across a variety of policy-relevant tasks, while requiring orders of magnitude less computing resources. In recent years, satellite imagery has become a key data source used to complement traditional surveys in data-scarce environments, allowing to predict with high accuracy cross-sectional variations in policy-relevant outcomes, from population density to the level of urbanization. However, a key roadblock in the wide-scale adoption of indicators derived from satellite images is that the barrier to entry is high, both in terms of computing resources as well as in well-trained computer scientists capable of analyzing them, especially in low-income countries where they arguably provide the most value. In this context, the method proposed by the authors could allow a wide variety of social scientists to derive insights from these images without having to directly analyse them. Beyond the methodological innovation described in the paper, the authors will make their features publicly available, which could have a high impact among development practitioners and for future research in this space.

I have 2 practical suggestions which could strengthen the results of the paper:

1- The authors argue that the MOZAIKS features only need to be computed and stored once. However, the physical characteristics measured in the images change over time, which could deteriorate the quality of the predictions produced by features computed using images from the past. I would encourage the authors to show the time-stability of the MOZAIKS features.

We appreciate this comment, as it reveals errors in our exposition. We apologize for the confusion, and have worked to clarify the claim and implications of the present analysis described in the main text.

The design of MOSAIKS requires that features need only be computed and stored once to predict multiple outcomes for each set of underlying imagery. However, if multiple time periods were being studied, this would require multiple sets of imagery, and thus multiple applications of the MOSAIKS featurization. Features computed from imagery in the past should not predict ground-level conditions if conditions on the ground have changed (as the reviewer correctly suggests). The language “computed and stored once” was in reference to many users all studying different task but within the same window of time.

In the manuscript, we limited our focus to a single set of imagery where we predicted variation across space for many tasks at a single point in time. We did not mean to suggest that imagery from the past should always capture all future events on the ground. As we are using the Google imagery base layer, we only have one set of imagery for each location. Detecting changes over time within a location using MOSAIKS is theoretically possible and it is a question we are investigating in ongoing work. However, it is beyond the scope of the present analysis since it requires obtaining/featurizing new sources of imagery that are available over time and developing new methods and tools to validate. We did not mean to imply in the manuscript that featurizing an image only once would be sufficient to monitor changes over time and apologize for the confusion.

To address this concern and prevent similar confusion, we have amended text in lines 36-37 and 58-60 in the introduction and in lines 72-74, where we describe the featurization process in more detail, so as to clarify for future readers what we mean by “features need only be computed and stored once”:

Lines 36-37: “We focus here on the problem of predicting properties of small regions (e.g. average house price) **at a single time period**, using high-resolution daytime satellite imagery as the only input.”

Lines 58-60: “Our approach allows common sources of imagery to be converted into **centralized sets of features** that can be accessed by many researchers, each solving different tasks.”

Lines 72-74: “**For each image, the** unsupervised featurization step can be centrally executed once, producing one set of outputs that are used to solve many different tasks through repeated application of the regression step by multiple independent users”

We are hopeful that, in the future, a system may exist where imagery is collected at regular time intervals, and at each interval, a new batch of MOSAIKS features is computed from the new images. Then, a user with time-dependent labels could match the appropriate time-stamped features with their time-stamped labels, up to a reasonable temporal frequency. However, demonstrating that such a system is feasible and accurate is a major undertaking, requiring use of data from an alternative satellite system, at least an order-of-magnitude more data, and extensive development of entirely new methods and tools to validate. Thus, we felt it was substantially beyond the scope of the present analysis and would require a dedicated study to evaluate satisfactorily. For example, roughly half of the analysis presented in Yeh et al (2020, *Nature Communications*) was dedicated to evaluating the performance of a CNN-based system over time. Given the space constraints of the journal, and the already long nature of this text, it did not seem feasible to include a similar analysis in this manuscript. However, we are currently working to answer the question posed by the reviewer in ongoing work in an independent study using entirely new data. To make clear that we have not yet achieved this goal in the present study, we have amended text in lines in the 329-332 discussion to explicitly state that a future system we envision could provide features over multiple time steps so that this is clear for future readers:

Lines 329-332: “We hope that MOSAIKS lays the foundation for **the future development of** an accessible and democratized system of global information sharing, where, **over time**, imagery from all available global sensors is continuously encoded as features and **appended to** a single table of data, which is distributed and used planet-wide.”

2- While the authors provided an extensive comparison of their approach with other SIML approaches, it would be interesting to show whether using the MOZAIKS features as additional features could help improve the predictive accuracy of the other SIML models they analysed.

To address this question, we developed a “hybrid” prediction model that incorporated MOSAIKS features into existing approaches. We then compared these results to MOSAIKS and other SIML benchmarks.

To test whether the addition of MOSAIKS features could improve performance of alternative SIML approaches, we focus on augmenting the best-performing alternative model (as seen in Fig. 3A): the end-to-end trained ResNet-18 CNN. We extract the 512 features generated by the last hidden layer of this model. As discussed in the manuscript, the linear regression step of MOSAIKS allows us to easily concatenate multiple feature sets, be they from other sensors (as demonstrated in the Extensions section, lines 264-275) or, in this case, from alternative SIML models. We thus concatenate the 512 ResNet-18 features with the 8,192 MOSAIKS features used throughout the manuscript and implement a ridge regression on the aggregate feature set, allowing for independent regularization parameters on each of the two source feature sets. This approach allows the features from the CNN and MOSAIKS to inform predictions in a ridge regression, similar to how they contribute information to predictions from the two separate approaches. Thus, the hybrid approach should not perform worse than either model, although it is theoretically ambiguous whether the hybridization can improve performance relative to the best-performing of the two approaches.

Our results indicate negligible performance gains from combining CNN and MOSAIKS features for most tasks, but notable boosts for the most difficult tasks: income and housing price. These two tasks also are those with the lowest correlation between ResNet-predicted values and MOSAIKS-predicted values (as shown in Supplementary Materials Fig. S17). This finding, along with a brief interpretation, is now included in new text added to the results section (lines 175-185):

“These experiments additionally reveal that patterns of predictability across tasks are strikingly similar in MOSAIKS and in alternative SIML approaches (Figs. S16 and S17). Together, these findings are consistent with the hypothesis that there exists some performance ceiling for each task, due to some factors not being observable from satellite imagery. To investigate this further, we develop a hybrid model in which the 512 features produced by the last layer of the ResNet-18 CNN are concatenated with the 8,192 MOSAIKS features and included together in a ridge regression. Performance improvements above either MOSAIKS or the CNN are small ($\leq 0.01R^2$) for most tasks, although there is a notable performance boost for the two tasks where both models achieve the lowest accuracy ($R_{hybrid}^2 - R_{CNN}^2 = 0.04$ for income; $R_{hybrid}^2 - R_{MOSAIKS}^2 = 0.05$ for housing price; Table S7). These results suggest that for some tasks, combining MOSAIKS with alternative SIML models can enhance predictive accuracy.”

We have also added the results of this comparison to the new Table S7 (suggested by Reviewer 2) in the Supplementary Materials, along with additional methodological clarification (lines 1476-1492):

“To further investigate this hypothesis, we test the performance of a hybrid approach that combines MOSAIKS features with features recovered from the ResNet-18 CNN. To do so, we use the same ridge regression method from MOSAIKS; however, prior to running the regression, we concatenate the 512 features produced by the last hidden layer of the ResNet-18 to the 8,192 MOSAIKS features used throughout our analysis. In the ridge regression, we apply independent regularization parameters for each of the two feature sets, effectively allowing the model to rely more heavily on one or the other.

Table S7 shows support for the hypothesis that MOSAIKS and the ResNet-18 CNN reflect similar image characteristics, as we find only a minimal performance gain from this hybrid approach (third column) for most tasks. However, we do see greater performance gains for the lowest performing tasks (income and housing price), which are also the tasks with the lowest correlation between MOSAIKS and the ResNet-18 predictions (Fig. S17). Note that in Table S7, performance metrics for MOSAIKS and the ResNet-18 differ slightly for some tasks when compared to results in Table S5, as the test set was defined slightly differently.^a

The demonstrated performance boost on the lowest-performing tasks indicates promise for MOSAIKS “hybrid” type models suggested by the reviewer. We thank the reviewer for pointing us toward this analysis, which strengthened insights within the paper.

^aThe hybrid approach relies on features defined by the ResNet-18 CNN, which was trained on 80% of the data. However, this method must also use a validation set to tune the ridge regression hyperparameters. For this tuning, we extract half of the remaining 20% of the data typically used as a test set. Results are reported on the remaining, untouched, 10%. Because this test set is slightly different than that used for the individual methods in Table S5, performance can vary slightly. Performance of all methods shown in Table S7 are shown for the same 10% test set.

Reviewer #2

(original comments in gray and italicized, replies in black and indented,
new text added to manuscript in bold)

The major novelty of this work, from a methodological point of view, lays in the demonstration that a “universal” unsupervised feature extraction process can be applied to satellite image observations, which can then be utilized, in a cost-effective way, to estimate a diverse set of geo-bio physical parameters. Using as input satellite observations from Google Earth, the paper proposed a very detailed and extensive experimental analysis of the behavior of the proposed approach, including the ability to make predictions to new locations where imagery is available but ground truth data may not be. The paper is very well written, the motivation of this work is founded, the experimental analysis is very thorough, and the conclusions are justified.

An important issue regarding the evaluation protocol however is the known limitation when considering google map images, in that neighboring spatial locations (even within the 256x256 region considered here) in a single input image, could have been acquired at distinct, and in general unknown, time instances. On the other hand, “ground” labels are obtained during specific periods in time. How does the proposed approach account for the fact that the input (satellite imagery) and the predictions can be characterized by a varying degree of temporal affinity? Results however demonstrate that the performance, even under the image acquisition uncertainty, is impressive. If it is possible to obtain the time of imaging from Google images, then an analysis of the impact in terms of classification performance would be very convincing.

As the reviewer points out, imagery from the Google Static Maps API, which we use to compute random convolutional features, are not associated with meta-data describing the time at which that image was collected. Similarly, the labeled data we obtain for each task were originally collected from different points in time, given distinct data availability constraints. Thus, the analysis is matching “recent” imagery (according to Google) with the most recent publicly-available labels, and we are unable to precisely match the time period for which the task labels were collected. Both of these facts were originally detailed in Supplementary Materials Section S.2.

In response to the reviewer’s concern, we have revisited and investigated whether the time of collection for Google imagery can be identified. Unfortunately, it is not possible to obtain the time stamp(s) from the Google Static Maps API. This is a long-standing request of users of the API, as evidenced by this “issue”^a in Google’s public “issue-tracker”, which was opened in July 2014, was assigned to an engineer in June 2017, and been reassigned to nine different Google employees since. The issue remains unaddressed as of the most recent update in July 2020. Therefore, at this time, it is not possible for us to quantify the temporal mismatch between our imagery and labels, nor to assess its performance implications.

Given this limitation in the data, we view the performance reported in the paper as conditional on a certain degree of unknown temporal mismatch. We hypothesize that if labeled data and imagery could be collected at exactly the same moment in time, performance could possibly improve, with performance gains that would likely be larger for tasks where large changes occur over time. To test this idea, we are actively in the process of acquiring high-resolution satellite imagery from a another data provider where time-stamps are available in ongoing work. However, we believe that analysis is beyond the scope of the current manuscript, as it requires the development of numerous new tools and methods to validate performance over time, in addition to contracting for new data acquisition and validating MOSAIKS for that new data source. In our view, it will require a new full manuscript to present that analysis and results satisfactorily and in full.

Nonetheless, to address the reviewers concern and make clear this important limitation of the current analysis, we have added to the main text the following clarifying sentence (lines 138-142):

“Note that due to the absence of metadata describing the exact time of observation in the Google imagery, as well as task-specific data availability constraints, these performance measures are conditional on a certain degree of unknown temporal mismatch between imagery and task labels (Supplementary Materials Section S.2).”

^a<https://issuetracker.google.com/issues/35824769>

Comparison with alternative paradigms, e.g., pre-training CNNs on natural images, or using models from one task for another, presented in Section 4 of the supplementary material, could benefit from concrete performance metrics (potentially formatted in tables), in addition to figures like S16 or S17, which makes it difficult to quantify the absolute performance gains/losses of the proposed approach.

This is a helpful suggestion that will improve the readability of our paper by adding key results in tabular format in addition to visual representations. We have made the following additions to the paper in response:

We have added Table S5 to the Supplementary Materials (page 81), which provides out-of-sample R^2 performance metrics for each task for MOSAIKS, the ResNet-18 CNN, and a pre-trained CNN. These are the same R^2 values shown visually in Figure 3A in the main text, but are now additionally provided in tabular format in Supplementary Materials Section S.4. The new table is shown below (blue text indicates a new addition to the manuscript):

Task	MOSAIKS R^2	ResNet-18 R^2	Pre-trained CNN R^2
Forest cover	0.91	0.94	0.66
Elevation	0.68	0.80	0.32
Population density	0.72	0.80	0.29
Nighttime lights	0.85	0.89	0.48
Income	0.45	0.47	0.07
Road length	0.53	0.58	0.16
Housing price	0.52	0.50	0.01

Table S5: **Comparison of model performance between MOSAIKS, a fine-tuned ResNet-18, a pre-trained ResNet-152, and a hybrid MOSAIKS-ResNet18 model.** Task-specific MOSAIKS test-set performance (first column) in contrast to: an 18-layer variant of the ResNet Architecture (ResNet- 18) trained end-to-end for each task (second column); an unsupervised featurization using the last hidden layer of a 152-layer ResNet variant pre-trained on natural imagery and applied using ridge regression (third column).

We have also added Table S6 to the Supplementary Materials (page 85). This table reports mean out-of-sample R^2 values across five folds of cross-validation for all four prediction methods, three countries, and eleven tasks shown visually in Fig. S16. The new table is shown below:

Country	Task	Method			
		RCF	Nightlights	RCF+NL	Head et al.
Rwanda	Wealth	0.72	0.74	0.74	0.74
	Electricity	0.68	0.71	0.71	0.69
	Mobile Phone Ownership	0.48	0.51	0.53	0.55
	Education	0.44	0.48	0.48	0.47
	Bed net count	0.38	0.10	0.38	0.40
	Female BMI	0.26	0.25	0.29	0.37
	Water access	0.26	0.17	0.26	0.26
	Child height %ile	0.23	0.21	0.25	0.20
	Child weight %ile	0.04	0.11	0.11	0.13
	Hemoglobin level	0.05	0.00	0.05	0.07
	Child weight / height %ile	0.00	-0.02	0.01	0.06
Haiti	Wealth	0.45	0.51	0.51	0.51
	Electricity	0.50	0.58	0.59	0.54
	Mobile Phone Ownership	0.27	0.26	0.31	0.38
	Education	0.47	0.50	0.52	0.56
	Bed net count	0.06	0.02	0.06	0.05
	Female BMI	0.24	0.23	0.26	0.31
	Water access	0.12	0.23	0.23	0.25
	Child height %ile	0.00	0.06	0.06	0.02
	Child weight %ile	0.00	0.02	0.02	0.01
	Hemoglobin level	0.00	0.01	0.01	-0.01
	Child weight / height %ile	-0.01	-0.01	-0.01	-0.02
Nepal	Wealth	0.48	0.53	0.57	0.64
	Electricity	0.14	0.13	0.19	0.24
	Mobile Phone Ownership	0.26	0.25	0.33	0.43
	Education	0.32	0.32	0.37	0.48
	Bed net count	0.50	0.29	0.51	0.63
	Female BMI	0.25	0.16	0.29	0.32
	Water access	0.23	0.18	0.24	0.33
	Child height %ile	0.09	0.07	0.11	0.07
	Child weight %ile	0.05	0.03	0.07	0.18
	Hemoglobin level	0.21	0.14	0.21	0.34
	Child weight / height %ile	0.02	0.01	0.02	0.11

Table S6: **Comparison of accuracy between MOSAICS and a transfer learning model.** All columns report out-of-sample mean R^2 values, where averages are taken across five folds (ranges across all five folds are shown visually in Fig. S16). Prediction methods are the same as in Fig. S16, where “Head et al.” indicates the transfer learning model from ref. (13).

We have also added to Figure S17 (page 87) the squared correlation coefficient between CNN predictions and MOSAIKS predictions (column 1), and between CNN errors and MOSAIKS errors (column 2). The updated figure is shown below:

Figure S17: **Comparison of predictions and prediction errors between MOSAIKS and the ResNet-18 CNN.** The left column shows the relationship between predictions generated by MOSAIKS (x -axis) and predictions generated by the ResNet-18 CNN (y -axis). The right column shows the relationship between prediction errors from MOSAIKS (x -axis) and prediction errors from the CNN (y -axis). In both plots, each point indicates one grid cell ($\sim 1\text{km} \times 1\text{km}$) in the holdout test set; the test set sample size is approximately 20,000 for each task, although sample sizes vary somewhat due to data availability across tasks (Section S.3.5). ρ^2 values on each plot indicate the square of the Pearson correlation coefficient.

Note that these changes include tabular and/or quantitative results associated with Figure 3A in the main text and all figures in Supplementary Materials Section S.4, except for Figure S15, which compares performance between MOSAIKS and GIST features for one task (housing price) in one region (Arizona, US). This figure is reproduced from a previous publication, which provides detail on the quantitative comparison across these methods.

Another aspect of the analysis which could also benefit from more concert result involved the process of “label super-resolution”. In this case, a plot showing the error (e.g. R^2) with respect to the super-resolution factor would be ideal.

We believe a plot corresponding to the reviewer’s request was included in the the original manuscript, although we understand it could be easily missed given the length of the manuscript.

Figure S13 plots R^2 values against super resolution factors:

Figure S13: Systematic evaluation of *within-image* R^2 recovered in the forest cover task.

The figure was also originally discussed in Supplementary Materials Section S.3.9 (lines 1275-1281):

“The resulting performance of label super-resolution predictions at different scales is shown in Fig. S13 for width scales of 2×2 , 4×4 , 8×8 , 16×16 , and 32×32 . We test up to $w = 32$ because the native width of the forest cover labels ($\sim 30\text{m}$) is just under $1/32$ the width of the original image ($\sim 1\text{km}$). Label super-resolution predictions are trained only on the aggregate label at the image-level. Nonetheless, as Fig. S13 shows, we are able to explain over 50% of the within-image label variations at 2×2 super-resolution, and over 30% of the variation using 32×32 super-resolution grids.”

To address the reviewer’s concern and prevent similar confusion, we have amended the main text to explicitly describe the nature of this plot when pointing interested readers to the Supplementary Materials (lines 285-288):

“Provided only a single label per image, MOSAIKS recovers substantial within-image signal when predicting forest cover in 4 to 1,024 sub-labels per label (within-image $R^2 = 0.54\text{-}0.32$, see Fig. S13 for a plot of performance against number of sub-labels and Supplementary Materials Section S.3.9 for methodological details).”

We display results for only forest cover because that is the only variable for which we have raw labels at fine enough resolution to quantify the accuracy of super-resolution predictions within each image.

Minor: in supplementary material, on section “label super-resolution experiment” you refer to Figure 4B, but I think you mean Figure 4C.

Thank you for catching this mistake; it has been fixed.

Reviewer #3

(original comments in gray and italicized, replies in black and indented,
new text added to manuscript in bold)

This paper presents a generalizable and accessible machine learning approach for diverse prediction tasks with satellite images, e.g., forest cover, house price, road length, etc. Authors mentioned that the proposed method shows the competitive performance with deep neural networks and even better in magnitude lower computational cost, scales globally, delivers label super-resolution predictions, and facilitates characterizations of uncertainty. However, the reviewer has the specific comments as follows.

1) In the introduction part, a detailed description of various satellite images is missing. For example, analyzing and discussing the advantages and disadvantages of different data sources are needed. This can help readers understand your points or motivations, e.g., why the proposed method is effective to be designed for satellite images and further for different applications, thereby understanding our environment.

As the reviewer rightly points out, there are many different sources of satellite imagery available. The objective of this manuscript was to test the hypothesis that a single encoding of satellite imagery could generalize across diverse prediction tasks. To test this idea, we selected high-resolution RGB imagery as a first data source because of its high resolution, global availability, clear interpretability, and prior performance in the literature (e.g., Jean et al., 2016). However, our selected imagery is by no means the only possible data source for such an experiment. In the manuscript we begin to test the performance of other imagery sources, such as nighttime lights, by themselves and when combined with features from RGB imagery. We find promising results (lines 264-275), though the optimal data source(s) for constructing a generalizable set of SIML features remains an open question. In ongoing research, we are testing the relative and combined performance of alternative satellite imagery data sources.

To address the reviewer’s concern and clarify our motivations for selecting the RGB imagery used throughout our analysis, we have added to the introduction the following description (lines 38-42):

“We use this imagery to test whether a single embedding can generalize across tasks because it is globally available from the Google Static Maps API at fine resolution, is geo-rectified and pre-processed to remove cloud occlusions, and has been found to perform well in SIML applications (Supplementary Materials Section S.2.2) (Jean et al. 2016, Head et al. 2017), though in principle other data sources could also be used (Zhu et al. 2018).”

Zhu et al. (2018) was a citation suggested by the reviewer (in their 4th comment below) and provides a review of the advantages and disadvantages of different satellite data sources for remote sensing. We believe this citation will help readers understand the broad diversity and relative strengths of available satellite data.

To further help readers understand the potential advantages of using additional data sources, we have also added a citation to the discussion where we describe how integrating more of the the diverse sources of satellite data could be an important direction for future research (lines 325-328):

“We conjecture that integrating new diverse data, from both satellite and non-satellite sources, may substantially increase the predictive accuracy of MOSAIKS for tasks not entirely resolved by daytime imagery alone; such integration using deep learning models is an active area of research (Hong et al. 2020).”

Hong et al. (2020) was a citation also suggested by the reviewer and is a useful example demonstrating how deep learning frameworks can be applied to different satellite imagery sources in combination, thereby providing a point of reference for contextualizing the fusion of daytime and nighttime imagery in our analysis.

2) *Authors claimed the proposed method is superior to the deep learning-based methods. However, there is a lack of theoretical proof and also experimental results are insufficient since only a few compared methods are investigated, which is relatively hard to support the opinion.*

We apologize for the confusion regarding the claims made in the paper. While there is theoretical motivation for applying random convolutional features to satellite imagery (as discussed in lines 88-95 and Supplementary Materials Appendix S.1), we agree with the reviewer that this does not prove that MOSAIKS is superior to deep learning methods, and thus we did not make that claim in the original manuscript. We also agree with the reviewer that the empirical results presented within the manuscript would not support the claim that MOSAIKS is superior to deep learning methods, and we worked to ensure this claim was not made in the original manuscript.

We believe that the results we show support our claim that MOSAIKS “achieves accuracy competitive with deep neural networks at orders of magnitude lower computational cost”, as written in the abstract and referenced by the reviewer in the first paragraph of their comments.

Nonetheless, to address this concern and ensure that there is no remaining confusion, we have updated the text in the discussion to further clarify that there are many cases where the performance of deep learning methods will be worth their higher cost (lines 301-304):

“While we have shown that in many cases MOSAIKS is a faster and simpler alternative to existing deep learning methods, there remain contexts in which custom-designed SIML pipelines will continue to play a key role in research and decision-making, **such as where resources are plentiful and performance is paramount.**”

Additionally, we have carefully reviewed and clarified how we discuss the costs and benefits of MOSAIKS and alternative methods throughout the manuscript. For example, we have checked that we always write that MOSAIKS achieves performance “competitive with” deep learning methods (occurring multiple times throughout the text: see abstract and lines 34, 100, 144, 286, 1372) and checked that we do not describe MOSAIKS as superior to deep learning-based methods anywhere in the manuscript.

We also note that our Results section continues to explicitly describe the relative performance of these two approaches, showing empirically that MOSAIKS sometimes outperforms and sometimes underperforms compared to a deep-learning approach, for example in lines 148-150:

“We find that MOSAIKS exhibits predictive accuracy competitive with the CNN for all seven tasks (mean $R_{CNN}^2 - R_{MOSAIKS}^2 = 0.04$; smallest $R_{CNN}^2 - R_{MOSAIKS}^2 = -0.03$ for housing; largest $R_{CNN}^2 - R_{MOSAIKS}^2 = 0.12$ for elevation).”

3) *The reviewer is wondering how about the efficiency (running time and computational cost)? since the authors mentioned the proposed method is more faster and efficient.*

We apologize if these values were difficult to find in the original manuscript. We previously documented the run time and computational cost of MOSAIKS in the main text (lines 131-135):

“Computing the feature matrix \mathbf{X} from imagery took less than 2 hours on a cloud computing node (Amazon EC2 p3.2xlarge instance, Tesla V100 GPU). Subsequently, solving a cross-validated ridge regression for each task took 6.8 minutes to compute on a local workstation with ten cores (Intel Xeon CPU E5-2630) (Supplementary Materials Section S.4.2).”

We additionally compared this run time to the computational cost of the ResNet-18 CNN in the original main text (lines 146-153):

“This training took 7.9 hours per task on a cloud computing node (Amazon EC2 p3.xlarge instance, Tesla V100 GPU). We find that MOSAIKS exhibits predictive accuracy competitive with the CNN for all seven tasks (mean $R_{CNN}^2 - R_{MOSAIKS}^2 = 0.04$; smallest $R_{CNN}^2 - R_{MOSAIKS}^2 = -0.03$ for housing; largest $R_{CNN}^2 - R_{MOSAIKS}^2 = 0.12$ for elevation) in addition to being approximately 250 to 10,000 \times faster to train, depending on whether the regression step is performed on a laptop (2018 Macbook Pro) or on the same cloud computing node used to train the CNN (Fig. 3A, Supplementary Materials Section S.4.1 and Table S8).”

We also originally provided a detailed comparison between the run time and computational costs of MOSAIKS and a ResNet CNN in the Supplement Section S.4.2 “Comparing costs” (lines 1501-1544):

“In practice, high computational costs can limit the use of SIML methods – especially when resources are scarce, such as in government agencies of low-income countries (Haack et al. 2016) or research teams and NGOs with limited budgets. Specifically designed to address this challenge, MOSAIKS scales across many research tasks by decoupling featurization from task selection, model-fitting, and prediction. The computationally costly step of featurization is done centrally on a fast computer with a graphics processing unit (GPU); individual practitioners need only download the pre-computed features, merge on labels for the task they select, and run regressions. Because features are created and stored by a central entity, the research community makes use of a cached set of computations, reducing the overall computational burden of widespread SIML and any external social costs generated by these computations (Strubell et al. 2019). Additionally, this decoupling of task-agnostic computations from task-specific computations allows practitioners to run more diagnostic analyses on their tasks, such as those presented in Fig. 3 of the main text.

From the perspective of a user who can access pre-computed MOSAIKS features to train and validate a new task, we find that MOSAIKS is $\sim 250\times$ to $10,000\times$ faster than a state-of-the-art neural net architecture (ResNet), depending on the computational resources available to a MOSAIKS user (Table S7). Moreover, MOSAIKS performance is competitive with the ResNet on all tasks we have studied (Fig 3A). From the perspective of the entire computational ecosystem, which bears the cost of image featurization in addition to model training and testing, we find that MOSAIKS is $5.3\times$ faster than the ResNet when solving a single task. The relative efficiency of MOSAIKS grows with the number of tasks studied because MOSAIKS features can be reused across tasks.

For the ResNet, the times in Table S7 reflect our wall-clock time on a single Amazon EC2 instance for a single task, so that the time costs are similar to that of introducing a single new domain *ex post*. For MOSAIKS, Table S7 includes wall-clock times on three different computational platforms, as users may have access to different resources. We show times using the same GPU as we use for the ResNet comparisons, times on a local workstation with ten cores (Intel Xeon CPU E5-263), and times on a standard laptop (MacBook Pro). For both ResNet and MOSAIKS, we report in Table S7 model training time *after* using cross-validation to select optimal hyperparameters. For MOSAIKS, model training time on the local workstation with 10 cores is ~ 6.8 minutes when including cross-validation to select penalization parameters in ridge regression. The ecosystem-wide costs of featurization per task shown in Table S7 decline as MOSAIKS becomes more widely adopted, because features can be cached centrally and distributed without modification to multiple users who are training and/or testing SIML in common locations.

We considered only one CNN architecture, which we chose because of its use in previous remote sensing applications (Jean et al. 2016). We did not attempt to innovate in neural net architectural design or algorithms. While one could pursue targeted innovations in neural networks for remote sensing, such as in ref. (Zhong et al. 2017), we emphasize that our method is currently orders of magnitude faster for the user than off-the-shelf fine-tuned CNN methods (Table S7), does not require a GPU for prediction, and achieves competitive prediction performance (Fig. 3A). There is recent work that aims to train networks to learn a “common representation” that can generalize across tasks, but this is a subject of ongoing research (Ruder et al. 2017), requires the tasks to be known in advance, and has yet to be demonstrated or evaluated at scale.”

These results are also summarized in Table S8 (which remains from the original manuscript):

Component	ResNet Time (GPU)	MOSAIKS Time
Training set featurization ($N = 80k$)		~ 1.2 hours (GPU)
Model training	~ 7.9 hours	~ 2.8 seconds (GPU) ~ 50 seconds (10 cores) ~ 1.8 minutes (laptop)
Holdout set featurization ($N = 20k$)		~18 minutes (GPU)
Holdout set prediction	~ 40 seconds	< 0.01 seconds (GPU) ~ 0.1 seconds (10 cores) ~ 0.7 seconds (laptop)
Total cost to ecosystem	~ 7.9 hours	~ 1.5 hours (GPU)
Total cost to user	~ 7.9 hours	~ 2.8 seconds (GPU) ~ 50.1 seconds (10 cores) ~ 1.8 minutes (laptop)

Table S8: **Wall-clock times of components of MOSAIKS compared with a fine-tuned CNN.** Bold times are those that a practitioner using each method would incur (assuming MOSAIKS users have access to a standard laptop only). Model training time includes training *after* tuning for a single task for both ResNet and MOSAIKS. MOSAIKS was run using $K=8,192$ features. ResNet operations were run on an Amazon EC2 p3.2xlarge instance with a Tesla V100 GPU and 60GB of onboard RAM. Cost of computation on this machine is roughly \$3/hr. MOSAIKS operations are shown for runs on this same GPU, a local workstation with ten cores (Intel Xeon CPU E5-263), and a standard laptop (MacBook Pro).

4) In recent years, lots of advanced deep learning methods and benchmark datasets have been developed. The reviewer suggests discussing and analyzing these advanced and latest methods and also adding more experimental results to show the superiority of the proposed method, such as “More diverse means better: Multimodal deep learning meets remote-sensing imagery classification. *IEEE Transactions on Geoscience and Remote Sensing*, 2020”, “Deep learning for remote sensing image classification: A survey. *Wiley Interdisciplinary Reviews: Data Mining and Knowledge Discovery*, 2018, 8(6), e1264.”, “Comprehensive survey of deep learning in remote sensing: theories, tools, and challenges for the community. *Journal of Applied Remote Sensing*, 2017, 11(4), 042609.”, and “A survey on deep learning-driven remote sensing image scene understanding: Scene classification, scene retrieval and scene-guided object detection. *Applied Sciences*, 2019, 9(10), 2110.”

We thank the reviewer for these useful references. After reading them carefully, we found that their conclusions provide additional support for our selection of both a fine-tuned CNN and a pre-trained unsupervised CNN as benchmark algorithms in the SIML literature with which to compare performance of MOSAIKS.

Specifically, the review articles – Ball et al. (2017), Li et al. (2018), and Gu et al. (2019) – provide valuable motivation for our comparison between MOSAIKS and a fine-tuned CNN by stating that: “the leading model in deep learning is that of CNNs” (Li et al. 2018); that “the CNN is the most popular and published to date” (Ball et al. 2017); and that CNNs are the “milestone technique for abstracting the visual content of remote sensing image scenes” (Gu et al. 2019). Li et al. (2018) and Gu et al. (2019) additionally note that while unsupervised CNN features generally perform worse than fine-tuned (supervised) CNN features, the outputs of a pre-trained CNN have been used and built upon to achieve “remarkable results on scene classification of remote sensing images” (Li et al. 2018). This adds justification for our choice of a pre-trained CNN as a comparable unsupervised featurization method in our experimental results (Fig. 3A and Supplementary Materials S.4.1). While comparing the performance of MOSAIKS to that derived from all methods proposed in the SIML literature would be infeasible in this manuscript, we believe that the experimental results we present provide a robust and extensive evaluation of MOSAIKS based on the summary of the literature provided in these references. By demonstrating that the performance of MOSAIKS is competitive with the leading methods in the literature identified by these reviews, we can, by transitivity, conclude that MOSAIKS is also competitive with additional approaches that have lower performance.

To address the reviewer’s comment, we now clarify our motivations for selecting the benchmark algorithms we analyze in the manuscript, we now include these citations when motivating the comparisons between MOSAIKS and other leading methods on lines 144-145:

“First, we retrain end-to-end a commonly-used deep convolutional neural network (CNN) architecture (He et al. 2016, Li et al. 2018, Gu et al. 2019)...”

and lines 154-155:

“Second, we apply “transfer learning” (Pan et al. 2010) using the ResNet-152 CNN pre-trained on natural images to featurize the same satellite images (Li et al. 2018, Gu et al. 2019).”

We also now cite the review by Ball et al. (2017), which includes a section on how “practitioners and researchers have a potentially steep learning curve to create custom deep learning solutions” when discussing the high barriers to entry for SIML in our manuscript on lines 21-25.

We thank the reviewer for these valuable references, which have substantially strengthened the manuscript.

REVIEWERS' COMMENTS

Reviewer #1 (Remarks to the Author):

Thank you for taking the time to make these changes. I do not have any remaining suggestion or concern.

Reviewer #2 (Remarks to the Author):

The authors have convincingly responded to all issues raised by the reviewers both verbally and with additional experimental analysis when necessary. I thus recommend the paper is published.

Reviewer #3 (Remarks to the Author):

The authors have well addressed the reviewer's concerns. No more comments.